# Chemical Characteristics and Controlling Factors of Shallow Groundwater in the Lower Reaches of Changhua River Basin, Hainan Island, China

Dun Wang [1,2], Lizhong Zhang [1,*], Lixin Pei [2,*], Xiwen Li [2], Yamin Yang [1], Zeheng Chen [2] and Linde Liang [1]

[1] Command Center of Natural Resources Comprehensive Survey, China Geological Survey, Beijing 100055, China; wangdu20@mails.ucas.ac.cn (D.W.); yangyamin@mail.cgs.gov.cn (Y.Y.)
[2] Haikou Marine Geological Survey Center, China Geological Survey, Haikou 571100, China; lxw1818168@163.com (X.L.)
* Correspondence: zhanglizhong1015@126.com (L.Z.); peilixin1983@163.com (L.P.)

**Abstract:** In order to ensure a sustainable water resource for residents of the lower reaches of the Changhua River on Hainan Island, it is crucial to understand the chemical characteristics and formation mechanisms of their shallow groundwater. To achieve this, we collected and analyzed 100 groundwater samples using various methods. Our findings indicate that the groundwater is generally near-neutral to weakly alkaline, suggesting an oxidized environment. We identified 56 chemical types, with Ca•Na-HCO$_3$ and Ca-HCO$_3$ being the most common. Water–rock interactions—particularly with silicate rocks—are the primary factor influencing the water's chemical characteristics, with Na$^+$ and K$^+$ coming from the leaching of silicate rocks, and Ca$^{2+}$ and Mg$^{2+}$ originating from the dissolution of carbonate rocks. Unfortunately, human activities such as agricultural practices and domestic sewage have had a significant impact on the groundwater, with NO$_3^-$ levels surpassing SO$_4^{2-}$ in most samples. These findings provide valuable insights into the development and protection of the shallow groundwater in this area.

**Keywords:** Hainan Island; hydrogeochemistry; rock weathering; controlling factors; groundwater





## 1. Introduction

The chemical composition of groundwater is not only an important part of the hydrogeochemical cycle, but is also closely related to biophysical and chemical processes related to the presence of organic matter. Therefore, the study of groundwater chemistry is closely related to the ecological environment. [1–3], while the changes in the chemical environment of groundwater are closely associated with ecological stability and water safety [1,4,5]. Research on the chemical characteristics and formation mechanisms of groundwater is helpful in understanding the change processes of the groundwater environment and facilitating the scientific and rational protection and development of groundwater resources [6,7]. Therefore, exploring the chemical characteristics of groundwater and its controlling factors can not only reveal the influence of various factors on groundwater, but also have great significance for the development and utilization of groundwater resources, ecological environment protection, and the construction of an ecological civilization.

The chemical characteristics of groundwater are affected by natural factors such as precipitation, evaporation, infiltration surface water, and the sedimentary environment, as well as human factors such as pollution and mining, which are the product of long-term interactions between groundwater and the surrounding environment [6,7]. Scholars have used mathematical statistics, the Piper three-line diagram, the Gibbs diagram, the ion proportional coefficient method, correlation analysis and other analytical methods to study the hydrochemical characteristics of major watersheds in the world and their relationship with regional hydrogeology and rainfall climate conditions, revealing their formation processes and evolution model [8,9].

Hainan Island has been employed as a pilot ecological civilization zone and a pilot free trade zone, and under such a context, the demand for water resources for socio-economic development is increasing, and the groundwater supply is becoming more and more important [10,11]. The lower reaches of the Changhua River are located in the southwest of Hainan Island. Shallow groundwater in the lower reaches of the Changhua River is an important source of drinking, agricultural irrigation, and industrial water for residents living at both sides of the river, and it is also an important guarantee for the sustainable development of an ecological environment. However, affected by a series of factors such as human activities, climate change, and seawater intrusion, the groundwater environment in this area is facing severe challenges [12–16]. At the same time, groundwater also reacts chemically with rocks, which leads to great changes in the chemical composition of the water, showing different characteristics [17]. All of these problems have led to the urgent need to identify the groundwater quality problems, in order to effectively serve the ecological green development of Hainan Island. Therefore, this study comprehensively investigates the chemical characteristics of shallow groundwater in the lower reaches of the Changhua River, its formation mechanisms, dynamic changes, influencing factors, and solute sources through hydrogeological surveys, water sampling, hydrochemical analysis, and multivariate statistics in the study area. This study aims to provide a scientific reference for the protection and rational development of groundwater resources in the lower reaches of the Changhua River [10].

## 2. Study Area

### 2.1. Overview of the Study Area

The study area is located in the southwest of Hainan Island, China, and its administrative division covers six towns including Sigeng, Changhua, Haiwei, Shiyuetian, Wulie, and Sanjia. The geographic coordinates are 108°37′–108°59′ E and 19°9′–19°30′ N, covering about 900 km$^2$, as shown in Figure 1. The study area has abundant water resources. The region experiences a distinct tropical monsoon climate. Rainfall distribution is uneven, with more rainfall in the mountainous areas than in coastal areas [18,19]. The average annual rainfall is 1150 mm. There are evident dry and wet seasons throughout the year, with the wet season occurring from May to October, during which rainfall can account for 70–90% of the annual total. There is a significant spatial variation in evaporation within the region, increasing from the foothills to the coastal areas, with an average annual evaporation of about 2419 mm. Additionally, this area is a major production region for tropical cash crops on Hainan Island, which leads to significant water consumption [18,19].

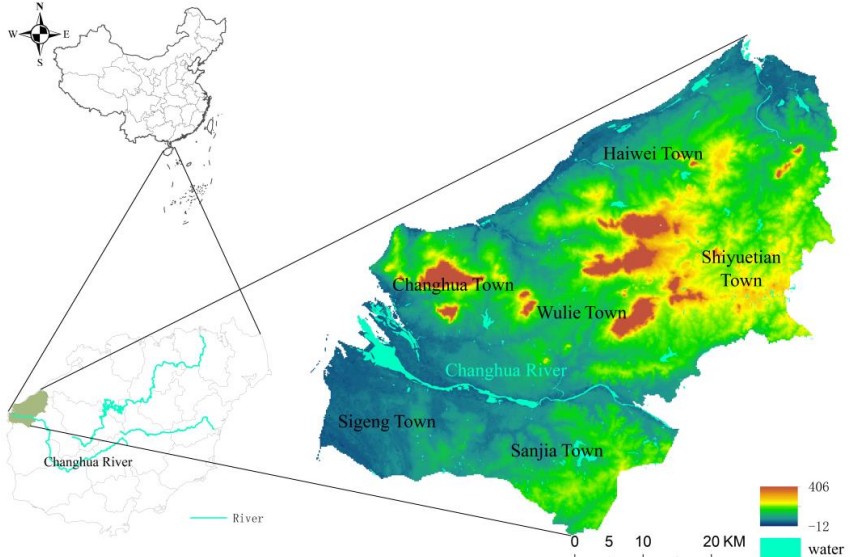

**Figure 1.** Location and digital elevation model (DEM) of the study area.

*2.2. Hydrogeological Conditions of the Study Area*

According to Figure 2, the groundwater in the study area is divided into two main categories: Quaternary loose rock pore water and bedrock fissure water. The loose rock aquifer is primarily distributed in the coastal accumulation layers, river alluvial–proluvial layers, and piedmont denudation accumulation layers. The coastal accumulation layers include sand bar terraces and sea terraces. In the sand bar terraces, the thickness of the rock layers is generally within 20 m, and the aquifer thickness is usually 5–15 m [20]. The lithology consists mainly of fine and medium sand with shells, pebbly sandy loam, medium coarse sand, and sandy gravel. The burial depth of the water level is generally less than 2 m, with a good water abundance. The sea terraces have an aquifer thickness ranging from 2 to 12 m. The river alluvial–pluvial terraces are distributed on both sides of the river, and the lithology of the aquifer mainly consists of pebbly sandy loam, pebbly medium-coarse sand, medium coarse sand, and sandy gravel. From the outer edge to the inner edge of the terrace, the aquifer thickness increases, sorting becomes better, and the water abundance is improved [21,22]. The bedrock fissure water is mainly distributed in the Changhua Daling and Sanjialing mountainous areas, and its water abundance is relatively poor. Atmospheric precipitation is the main recharge source for groundwater in the study area. Due to the loose formation lithology and good permeability, irrigation water is also an important recharge source in some areas. Groundwater runoff and discharge are controlled by the topography. The coastal sand bar terraces and the piedmont alluvial layers have a loose lithology and a certain slope, leading to favorable conditions for groundwater runoff and discharge. However, the river terraces and sea terraces have very small hydraulic gradients, resulting in relatively poor conditions for groundwater runoff and discharge. Groundwater discharge mainly occurs in low-lying areas such as gullies or at the edges of terraces, where it is released in the form of springs or sheet flows, eventually flowing into a gulley or the sea. In the coastal plain areas, the groundwater level is shallow and the evaporation is intense, with annual evaporation exceeding rainfall, making evaporation one of the methods for groundwater discharge. In the Piedmont areas, the groundwater level is generally higher than the confined water level, and the groundwater laterally recharges the confined water [21,22].

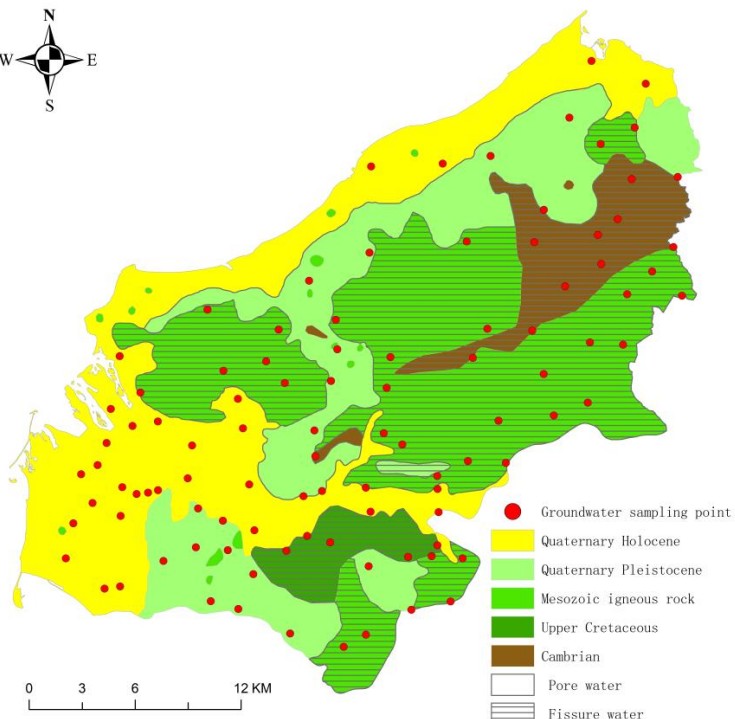

**Figure 2.** Schematic diagram of the distribution of sampling points.

## 3. Materials and Methods

### 3.1. Sample Collection

To investigate the hydrochemical characteristics and controlling factors of the lower Changhua River basin in Hainan Island, this study takes into account the hydrogeological conditions, geological and geomorphic features, as well as land use types in this area. The groundwater samples that can to the greatest extent represent the groundwater characteristics of the lower Changhua River basin in Hainan Island were collected. A total of 100 groundwater samples were collected, including 51 pore groundwater samples and 49 fissure groundwater samples, at well depths ranging from 5 m to 120 m. The distribution of the sampling points is shown in Figure 2.

Groundwater samples were collected using water pumps. For actively used groundwater wells, samples were directly collected. However, for abandoned or long-unused domestic wells, they were first cleaned before sampling. Additionally, water samples with a turbidity greater than 3NTU were filtered using a 0.45-micron membrane filter. All samples were stored at a low temperature (4 °C) and sent to the laboratory for analysis. During the sampling process, detailed data such as coordinates, well depth, and water level were recorded. The collection of samples was conducted under the guidance of experienced laboratory professionals. Before collecting samples, the sample bottles were soaked in dilute nitric acid and then rinsed with deionized water to ensure cleanliness. The sampling process involved rinsing the sample bottles 2–3 times with water samples before filling them with groundwater.

### 3.2. Sample and Data Processing

In this study, we selected various water sample parameters for analysis, including pH, dissolved oxygen (DO), oxidation-reduction potential (ORP), total dissolved solids (TDS), total hardness (TH), $Na^+$, $K^+$, $Ca^{2+}$, $Mg^{2+}$, $HCO_3^-$, $NO_3^-$, $Cl^-$, and $SO_4^{2-}$, etc. The analysis of $K^+$, $Na^+$, $Ca^{2+}$, and $Mg^{2+}$ was conducted using a flame atomic absorption spectrophotometer. $NO_3^-$, $Cl^-$, and $SO_4^{2-}$ were measured using an ion chromatograph. TDS was determined by a drying-weighing method, while the $HCO_3^-$ parameter was analyzed through titration. COD (chemical oxygen demand) was measured using a rapid closed-catalytic digestion method, and TH was determined by EDTA titration. The quality control for water sample testing was achieved by means of blank samples, parallel samples, and standard addition samples. The proportion of blank and parallel samples in each batch was 10–30%, and all blank samples showed no detection, while the parallel samples were within the acceptable range. The recovery rates of standard addition samples were between 86–113%, which fell within the acceptable range of 80–120%. The detection limits for each component are listed in Table 1.

**Table 1.** Detection limits for each component of the water sample.

| Component | Detection Limit/mg·$L^{-1}$ | Component | Detection Limit/mg·$L^{-1}$ |
|---|---|---|---|
| COD | 5 | $K^+$ | 0.07 |
| $HCO_3^-$ | 5 | $Na^+$ | 0.03 |
| $Cl^-$ | 0.007 | $Ca^{2+}$ | 0.02 |
| $SO_4^{2-}$ | 0.018 | $Mg^{2+}$ | 0.02 |
| $NO_3^-$ | 0.016 | | |

The data processing utilized Origin software for the descriptive statistical analysis of the groundwater analysis data in the study area, including the mean value, median value, standard deviation, minimum value, maximum value, and coefficient of variation. SPSS was used to analyze the groundwater chemical data with multivariate statistics. ArcGIS software was used to carry out spatial interpolation to investigate the spatial characteristics of the major ions in the shallow groundwater downstream of the Changhua River. Furthermore, a Piper diagram, Gibbs diagram, end-member diagram, and ion ratio

diagram were plotted to analyze the chemical characteristics and controlling factors of the groundwater in the study area.

## 4. Results and Discussions

### 4.1. Statistical Characteristics of the Main Chemical Indicators of Water

From Table 2, the overall groundwater pH in the study area ranged from 6.6 to 8.0, indicating its near-neutral to weakly alkaline characteristics. DO ranged between 2.52 and 7.22 mg/L and ORP ranged between 33.5 and 101.60 mV, with the mean values of 3.64 mg/L and 71.83 mV, respectively—indicating that the groundwater in the study area was generally in an oxidizing environment. The median value of TDS was 281.50 mg/L, with a range of 40–958 mg/L, indicating that the groundwater in the study area was freshwater and might not have suffered from the impact of seawater intrusion. The TH ranged between 0.22 and 4.87 mmol/L, and the mean value was 1.66 mmol/L. The median values of macro cations in the groundwater in the study area were ranked as $Ca^{2+}$ > $Na^+$ > $Mg^{2+}$ > $K^+$, and the median values of macro anions were ranked as $HCO_3^-$ > $Cl^-$ > $NO_3^-$ > $SO_4^{2-}$. Among these, the relatively large coefficients of variation for the $Ca^{2+}$, $Mg^{2+}$, $HCO_3^-$, and $NO_3^-$ indicated that these four ions have stronger spatial dispersion. It is important to note that $NO_3^-$ has become a macroscopic component exceeding $SO_4^{2-}$ in most of the sampling groundwater, indicating that human activities in the study area have a greater impact on groundwater [23,24].

**Table 2.** Statistics of characteristic values of chemical indicators of groundwater in the study area.

| Eigenvalue | pH | DO | ORP | TDS | TH | $K^+$ | $Na^+$ | $Ca^{2+}$ | $Mg^{2+}$ | $Cl^-$ | $HCO_3^-$ | $SO_4^{2-}$ | $NO_3^-$ |
|---|---|---|---|---|---|---|---|---|---|---|---|---|---|
| Minimum value | 6.61 | 2.52 | 34 | 40 | 0.22 | 0.3 | 5 | 4 | 3 | 4 | 6 | 1 | <DL |
| Maximum value | 8.01 | 7.22 | 102 | 958 | 4.87 | 120 | 274 | 129 | 634 | 342 | 585 | 281 | 163 |
| Median value | 7.17 | 3.24 | 74 | 282 | 1.49 | 8.4 | 32 | 37 | 16 | 39 | 103 | 28 | 31 |
| Mean value | 7.15 | 3.64 | 72 | 334 | 1.66 | 16.6 | 46 | 43 | 18 | 62 | 145 | 40 | 44 |
| Coefficient of variation | 1 | 0.89 | 1.03 | 0.84 | 0.89 | 0.5 | 0.69 | 0.85 | 0.87 | 0.63 | 0.71 | 0.7 | 0.71 |

Note: In the table, ORP is in mV, TH is in mmol/L, the rest of the water chemical indicators are in mg/L, and the coefficient of variation is in %.

### 4.2. Chemical Types of Groundwater

As shown in Figure 3. The chemical conditions of the groundwater in the study area were complex, with as many as 56 chemical types of groundwater. The Ca•Na-$HCO_3$ type of water accounted for the largest share of 10%, followed by the Ca-$HCO_3$ and Na•Ca-$HCO_3$•Cl types—both accounting for 7%—while the remaining 53 chemical types all accounted for less than 5%. By grouping and simplifying the chemical types and retaining only one of the most dominant cations and anions, the groundwater chemical types were simplified from 56 to 11 [25]. Among them, the Ca-$HCO_3$ type water dominated with 37%, followed by the Na-$HCO_3$ type water (21%) and the Na-Cl type water (11%), respectively, and the remaining eight chemical types were all within 10%. It is worth mentioning that the $NO_3$-type water accounted for 11%, indicating a more pronounced influence of human activities [26].

In the study area, the $HCO_3$-type water accounted for over 60% of the total area and was mainly distributed in Haiwei Town, Shiyuetian Town, and the adjacent area of Changhua, Sigeng, and Sanjia Town. The $SO_4$-type water ranked second, accounting for approximately 20% of the area. It was primarily distributed in the northern part of Changhua Town, the adjacent area of Wulie, Sanjia, Shiyuetian Town, the western and southern parts of Sigeng Town, and the central-eastern part of Haiwei Town. The Cl-type water and $NO_3$-type water each accounted for less than 10% of the area. Specifically, the $NO_3$-type water was mainly distributed in the urban areas adjacent to Wulie, Sanjia, and Shiyuetian Town, indicating a close correlation with human activities such as urbanization in the study area [5,27].

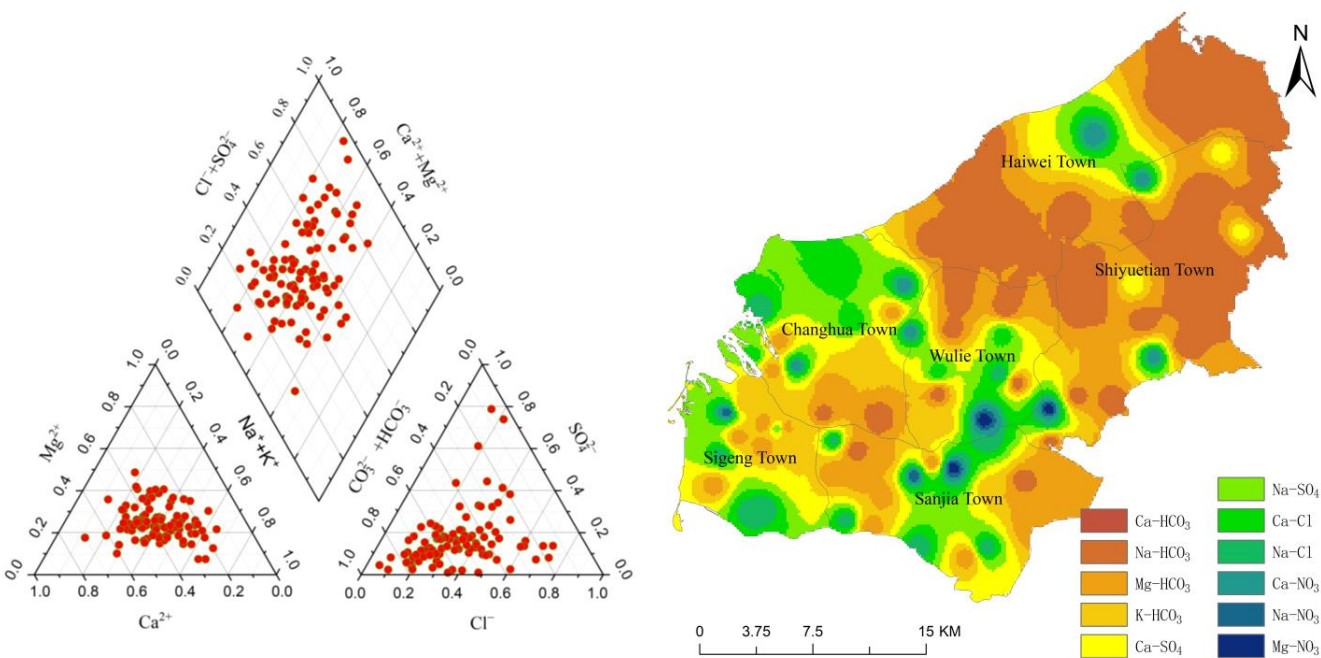

**Figure 3.** Chemical types and spatial distribution of groundwater in the study area.

*4.3. Analysis of Groundwater Chemical Controlling Factors*

4.3.1. Water–Rock Model Analysis

The study area is located in a tropical climate zone and is greatly influenced by seasonal rainfall. Both the infiltration of atmospheric rainfall and river water runoff contribute to groundwater recharge [18,19,23]. Additionally, reactions between groundwater and the rock formations in the aquifers can also cause chemical changes in groundwater. Therefore, it is essential to explore the groundwater chemical characteristics and the controlling factors in the study area [28]. Gibbs diagrams are commonly used to study the impact of water–rock interactions on groundwater chemistry; they provide a macroscopic understanding of the controlling factors for the major ions in groundwater, including atmospheric precipitation, rock leaching, and evaporation–crystallization [24,29,30].

As shown in Figure 4a,b, the ratio of cation mass concentration $Na^+/(Na^+ + Ca^{2+})$ ranged from 0.20 to 0.80, and the ratio of anion mass concentration $Cl^-/(Cl^- + HCO_3^-)$ ranged from 0.10 to 0.80. Water samples with moderate total dissolved solids and low ratios were located in the rock weathering control area, i.e., the middle-left part of the graph, indicating a rock weathering effect. This suggests that water–rock interactions dominate the hydrogeochemical processes in the study area. Some water samples fall within the region influenced by evaporation concentration, indicating a certain degree of impact from evaporation concentration. No water samples were located in the bottom-right region of the graph, which represents the area influenced by atmospheric precipitation. This suggests that atmospheric precipitation may have an influence, but its effect is not significant.

The $Na^+$ end-member method was used to further explore the influence of rock weathering on the groundwater hydrochemical evolution process in the study area. According to the concentration ratio of $(HCO_3^-/Na^+)/(Ca^{2+}/Na^+)$ and $(Mg^{2+}/Na^+)/(Ca^{2+}/Na^+)$, the main weathering sources of groundwater ions are divided into three types—namely, carbonate rock, silicate rock, and evaporite rock [31,32]. It can be seen from Figure 4c,d that the shallow groundwater samples from the Changhua River Basin were mainly concentrated in the middle of the silicate rock end-member, and a small number were biased toward the carbonate rock and evaporite rock control end-members, indicating that the groundwater in this area is mainly affected by silicate rock weathering, followed by carbonate rock and evaporite rock weathering.

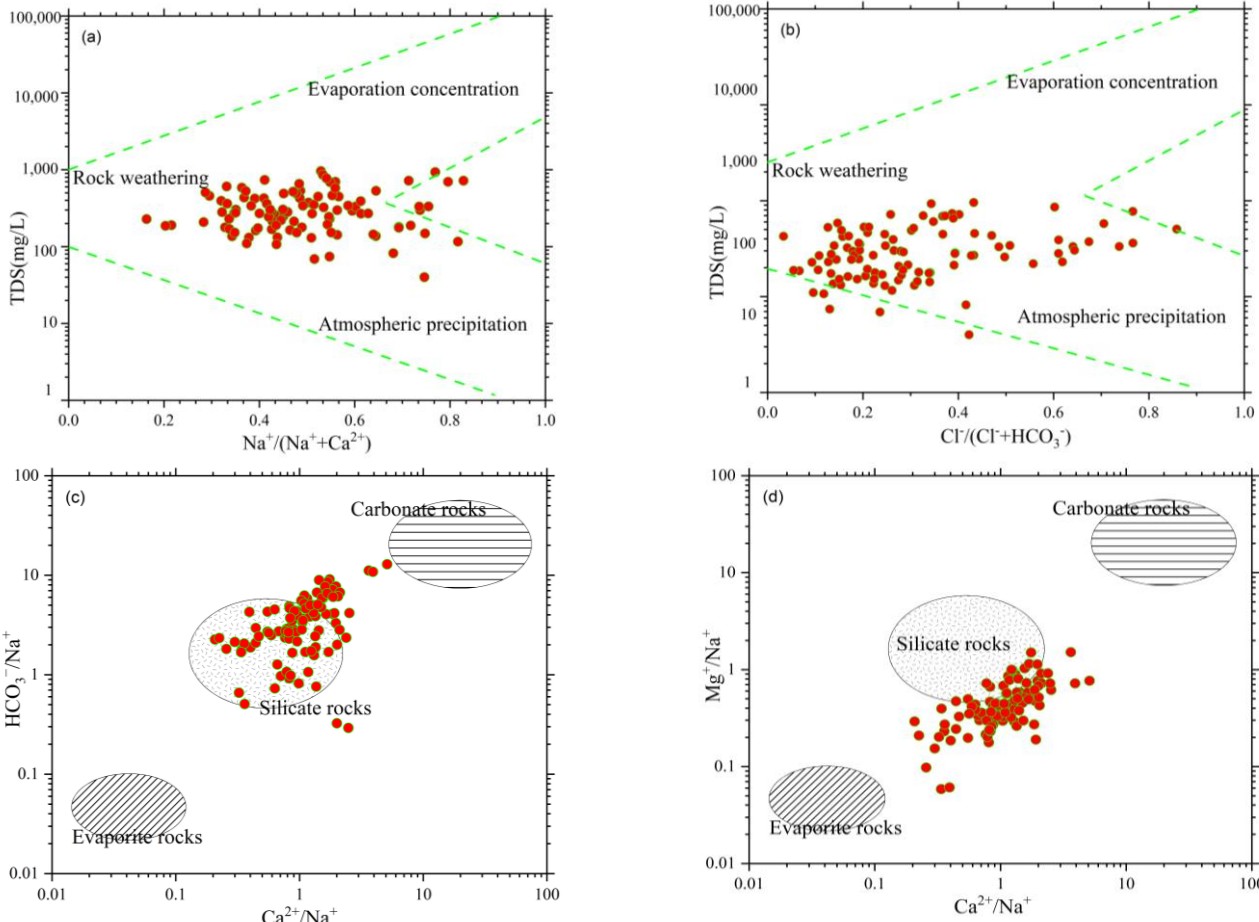

**Figure 4.** Gibbs diagram and $Na^+$ end-member diagram of groundwater in the study area: (**a**) relationship between $Na^+/(Na^+ + Ca^{2+})$ and TDS; (**b**) relationship between $Cl^-/(Cl^- + HCO_3^-)$ and TDS; (**c**) relationship between $Ca^{2+}/Na^+$ and $HCO_3^-/Na^+$; (**d**) relationship between $Ca^{2+}/Na^+$ and $Mg^{2+}/Na^+$.

### 4.3.2. Analysis of Ion Ratio Relationship and Source of Main Components

Previous research commonly used correlation analysis between various chemical components of groundwater to reveal their sources [33,34]. Based on the correlation analysis results of various chemical components in Table 3, TDS shows a significant positive correlation with $K^+$, $Na^+$, $Ca^{2+}$, $Mg^{2+}$, $Cl^-$, $HCO_3^-$, $SO_4^{2-}$, and $NO_3^-$ ($p < 0.01$). This indicates that these chemical components have a significant contribution to the TDS. Among them, the correlations between TDS and $Na^+$, $Ca^{2+}$, $Mg^{2+}$, and $Cl^-$ are the most significant, with correlation coefficients of 0.797, 0.842, 0.802, and 0.837, respectively. This suggests that these four components are the primary factors influencing the TDS [35].

$HCO_3^-$ shows significant correlations with $Na^+$, $Ca^{2+}$, and $Mg^{2+}$, indicating that $HCO_3^-$ shares a common source with $Na^+$, $Ca^{2+}$, and $Mg^{2+}$, possibly originating from the weathering and dissolution of silicate rocks or carbonate rocks. The significant correlation of $Na^+$ with $Cl^-$ and $HCO_3^-$ suggests the presence of weathering and the dissolution of sodium-bearing silicate minerals such as sodium feldspar, which could also come from atmospheric precipitation and the dissolution of evaporite rocks. The correlation coefficient of 0.83 between $Na^+$ and $Cl^-$ indicates a common source for both ions. The study area is close to the ocean, whereas the average $Cl^-/Na^+$ (mEq/L) ratio of 0.85 is lower than the world average seawater ratio ($Cl^-/Na^+ = 1.16$), suggesting that their chemical components may not be significantly influenced by seawater intrusion [33,36]. Furthermore, the correlation coefficient between $K^+$ and $Cl^-$ was 0.169, 0.421 between $K^+$ and $NO_3^-$ and 0.136 between $Cl^-$ and $NO_3^-$, respectively. Potassium chloride (KCl) is a common

source of $K^+$ and $Cl^-$, and potassium nitrate ($KNO_3$) is a common source of $K^+$ and $NO_3^-$, suggesting that a portion of $K^+$, $Cl^-$, and $NO_3^-$ could come from agricultural activities such as fertilizer usage and the discharge of domestic wastewater [37].

**Table 3.** Correlation coefficient matrix of chemical parameters of groundwater in the study area.

| | PH | DO | OPR | TDS | $K^+$ | $Na^+$ | $Ca^{2+}$ | $Mg^{2+}$ | $Cl^-$ | $HCO_3^-$ | $SO_4^{2-}$ | $NO_3^-$ |
|---|---|---|---|---|---|---|---|---|---|---|---|---|
| PH | 1 | | | | | | | | | | | |
| DO | −0.100 | 1 | | | | | | | | | | |
| OPR | −0.070 | 0.318 ** | 1 | | | | | | | | | |
| TDS | 0.052 | −0.013 | −0.060 | 1 | | | | | | | | |
| $K^+$ | −0.117 | 0.061 | 0.102 | 0.323 ** | 1 | | | | | | | |
| $Na^+$ | 0.207 * | 0.086 | 0.027 | 0.797 ** | 0.094 | 1 | | | | | | |
| $Ca^{2+}$ | 0.136 | −0.052 | −0.174 | 0.842 ** | 0.221 * | 0.545 ** | 1 | | | | | |
| $Mg^{2+}$ | 0.120 | −0.139 | −0.158 | 0.802 ** | 0.142 | 0.564 ** | 0.726 ** | 1 | | | | |
| $Cl^-$ | 0.137 | 0.120 | 0.012 | 0.837 ** | 0.169 | 0.834 ** | 0.686 ** | 0.598 ** | 1 | | | |
| $HCO_3^-$ | 0.372 ** | −0.116 | −0.062 | 0.655 ** | 0.160 | 0.700 ** | 0.604 ** | 0.665 ** | 0.515 ** | 1 | | |
| $SO_4^{2-}$ | −.270 ** | 0.123 | −0.011 | 0.567 ** | 0.255 * | 0.387 ** | 0.579 ** | 0.406 ** | 0.397 ** | 0.141 | 1 | |
| $NO_3^-$ | −0.155 | −0.024 | −0.047 | 0.295 ** | 0.421 ** | 0.099 | 0.179 | 0.258* | 0.136 | −0.083 | 0.155 | 1 |

Note: ** indicates a significant correlation at the 0.01 level (two-sided); * indicates a significant correlation at the 0.05 level (two-sided).

In this section, we further investigate the controlling factors of the chemical characteristics of the shallow aquifers downstream of the Changhua River. Due to the reaction between groundwater and aqueous media, the proportional relationship of various ions in groundwater is conducive to clarify the sources of major ions and hydrogeochemical processes. Therefore, a comparison of the main indicators of the 100 groundwater samples in the study area was conducted using milliequivalent ratios, as shown in Figure 5.

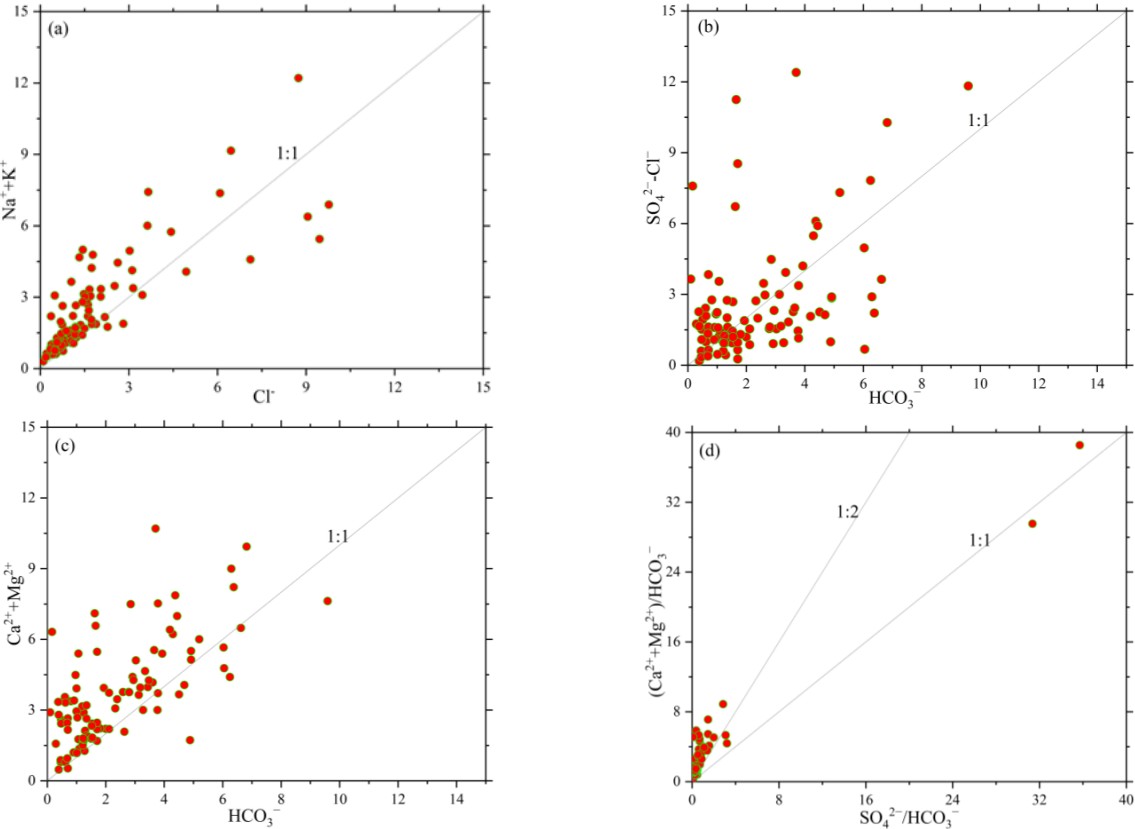

**Figure 5.** Proportional relationship of main ions: (**a**) relationship between $Na^+ + K^+$ and $Cl^-$; (**b**) relationship between $SO_4^{2-} + Cl^-$ and $HCO_3^-$; (**c**) relationship between $Ca^{2+} + Mg^{2+}$ and $HCO_3^-$; (**d**) relationship between $(Ca^{2+} + Mg^{2+})/HCO_3^-$ and $SO_4^{2-}/HCO_3^-$.

In general, the relationship between $n(Na^+ + K^+)$ and $n(Cl^-)$ in a groundwater system can provide insights into the sources of $Na^+ + K^+$ and $Cl^-$. For example, a ratio of $n(Na^+ + K^+)/n(Cl^-) = 1$ suggests dissolution from evaporite rocks, while a ratio of $n(Na^+ + K^+)/n(Cl^-) > 1$ indicates weathering from silicate rocks [6,38]. From Figure 5a, it can be observed that the majority of groundwater sampling points downstream of the Changhua River are located above the 1:1 line, indicating that the main source of $Na^+$ and $K^+$ in this area is likely from the leaching of silicate rocks. Meanwhile, some points are distributed below the 1:1 line, confirming that the groundwater chemical composition was influenced by the leaching of carbonate rocks. The ratio of $n(SO_4^{2-} + Cl^-)/n(HCO_3^-)$ reflects the dissolution of carbonate rocks in the study area. When $n(SO_4^{2-} + Cl^-)/n(HCO_3^-) > 1$, it suggests that $SO_4^{2-}$ and $Cl^-$ are derived from the weathering of evaporite rocks. Conversely, if $n(SO_4^{2-} + Cl^-)/n(HCO_3^-) < 1$, it indicates that $HCO_3^-$ is sourced from the weathering of carbonate rocks [39,40]. In Figure 5b, all the water sampling points are located above the 1:1 line, and some samples have much higher concentrations of $(SO_4^{2-} + Cl^-)$ compared to $HCO_3^-$, indicating that the groundwater was mainly influenced by the weathering of evaporite rocks such as gypsum, rock salt, and mirabilite.

The sources of $Ca^{2+}$ and $Mg^{2+}$ can be determined based on the milliequivalent concentration ratio of $(Ca^{2+} + Mg^{2+})/HCO_3^-$. When this ratio is greater than one, it indicates that $Ca^{2+}$ and $Mg^{2+}$ are primarily sourced from the dissolution of carbonate rocks. Conversely, if the ratio is less than one, it suggests that $Ca^{2+}$ and $Mg^{2+}$ are mainly derived from the dissolution of silicate rocks and evaporite rocks [41,42]. As shown in Figure 5c, most groundwater sampling points in the study area are located above the ratio line of one and only a small portion is located below, indicating that $Ca^{2+}$ and $Mg^{2+}$ in the groundwater primarily come from the dissolution of carbonate rocks. From the milliequivalent concentration ratios of $(Ca^{2+} + Mg^{2+})/HCO_3^-$ and $(SO_4^{2-}/HCO_3^-)$, we can further analyze the involvement of carbonic acid and sulfuric acid in the dissolution of carbonate rocks. When the ratio is two, it suggests that sulfuric acid participates in the dissolution process of carbonate minerals. When the ratio is 1:1, it indicates that carbonic acid is involved in the dissolution of carbonate minerals [41,42]. From Figure 5d, it can be observed that most water sampling points are located near the ratio line of 1:2 and the upper left corner, indicating that both carbonic acid and sulfuric acid in the water are involved in the dissolution of carbonate minerals, but the carbonic acid contributed significantly more than the sulfuric acid.

### 4.3.3. Analysis of Cationic Exchange Adsorption

Under certain conditions, the exchange of certain cations adsorbed on rock and soil surfaces with cations in groundwater is referred to as cationic exchange adsorption. It is commonly represented by the ratio of $n(Ca^{2+} + Mg^{2+} - SO_4^{2-} - HCO_3^-)/n(Na^+ + K^+ - Cl^-)$, where if cationic exchange occurs, the ratio of $n(Ca^{2+} + Mg^{2+} - SO_4^{2-} - HCO_3^-)/n(Na^+ + K^+ - Cl^-)$ is approximately equal to $-1$ [17,30]. In the case of the shallow groundwater in the lower reaches of Changhua River, there is a strong correlation between $n(Ca^{2+} + Mg^{2+} - SO_4^{2-} - HCO_3^-)$ and $n(Na^+ + K^+ - Cl^-)$, and the ratio is around $-1$ (Figure 6a)—indicating that cationic exchange is occurring in the study area.

The direction and strength of cationic exchange can be further represented by the Chloro–Alkali Index (CAI) [7,25]. Typically, when the $Ca^{2+}$ and $Mg^{2+}$ in groundwater undergo cationic exchange with the $Na^+$ and $K^+$ adsorbed on the surface of aquifer particles, both CAI-I and CAI-II are negative values. Conversely, if there is an anionic exchange process, CAI-I and CAI-II will be positive values.

From the relationship between CAI and TDS (Figure 6b), it can be observed that 81% of the water sampling points have negative CAI values. This indicates that the shallow groundwater in the study area primarily undergoes cationic exchange, where $Ca^{2+}$ and $Mg^{2+}$ in the pore water exchange with $Na^+$ and $K^+$ in the aquifer minerals, leading to a reduction in $Ca^{2+}$ and $Mg^{2+}$ concentrations and an increase in $Na^+$ and $K^+$ concentrations in the water.

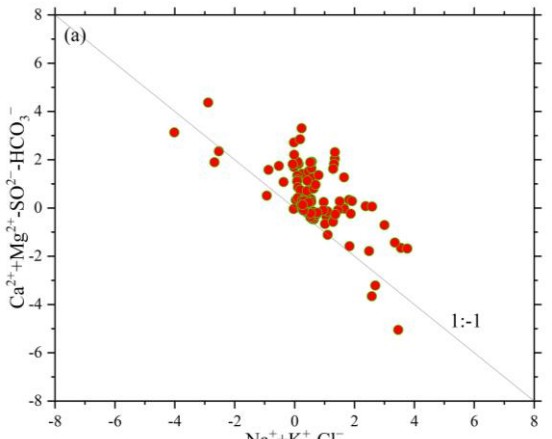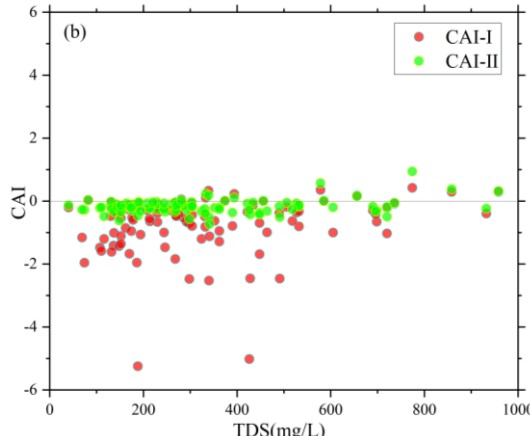

**Figure 6.** Cationic exchange adsorption: (**a**) relationship between $Ca^{2+} + Mg^{2+} - SO_4^{2-} - HCO_3^-$ and $Na^+ + K^+ - Cl^-$; (**b**) relationship between CAI and TDS.

### 4.3.4. Analysis of the Impact of Human Activities

In this study, the downstream area of the Changhua River in Hainan Island is the main tropical crop planting region with a wide rural area. Agricultural fertilization and untreated domestic sewage and waste can generate a large number of pollutants, such as chloride ions and nitrates, which can enter the shallow groundwater with rainwater or surface water—thereby affecting the hydrochemical evolution of the groundwater [10,11].

The higher the ratios of $Cl^-/Na^+$ and $NO_3^-/Na^+$ in groundwater, the more significant the impact of human activities [43]. From Figure 7a, we can observe that both ratios were relatively high, and some samples were close to or fell within the agricultural pollution end-member, indicating that the shallow groundwater had been influenced to some extent by agricultural pollution. The relationship between $SO_4^{2-}/Ca^{2+}$ and $NO_3^-/Ca^{2+}$ is commonly used to analyze the influence of human activities on the main ions in groundwater. When $SO_4^{2-}/Ca^{2+} > NO_3^-/Ca^{2+}$, it indicates a greater impact from industrial and mining activities, whereas the opposite suggests a greater influence from agricultural activities and domestic sewage; Figure 7b.

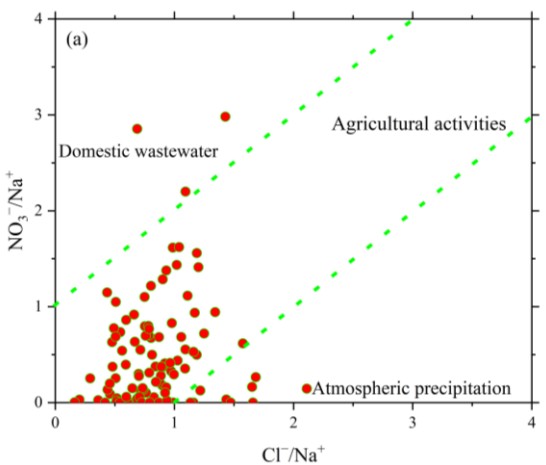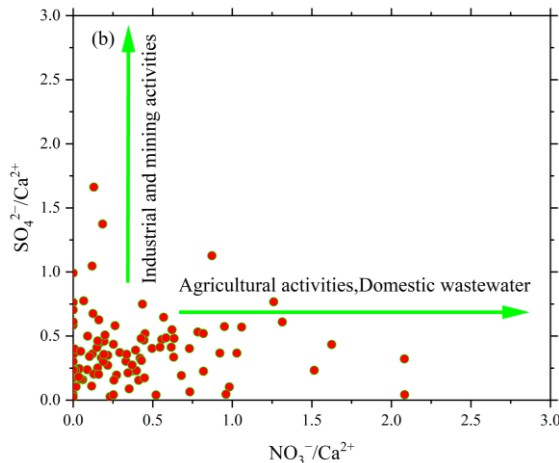

**Figure 7.** Impacts of human activities: (**a**) relationship between $NO_3^-/Na^+$ and $Cl^-/Na^+$; (**b**) relationship between $SO_4^{2-}/Ca^{2+}$ and $NO_3^-/Ca^{2+}$.

## 5. Conclusions and Outlook

This study comprehensively analyzed the hydrochemical characteristics of shallow groundwater in the downstream area of the Changhua River using multiple methods. The results showed that the overall pH of the groundwater in the study area ranged from 6.6 to

8.0, indicating a near-neutral to weak alkaline nature. The dissolved oxygen (DO) content ranged from 2.52 to 7.22 mg/L, with an average of 3.64 mg/L, and the oxidation–reduction potential (ORP) ranged from 33.5 to 101.60 mV, with an average of 71.83 mV—indicating an overall slightly oxidizing environment for the groundwater in the study area. The groundwater in the study area was classified as freshwater, suggesting that it might not be affected by seawater intrusion.

By using the Piper diagram and the Shukalev classification method, the chemical types of groundwater in the study area were classified into 56 types. The most abundant types were the Ca•Na-$HCO_3$ type, the Ca-$HCO_3$ type, and the Na•Ca-$HCO_3$•Cl type. The $NO_3$-type water was mainly distributed in the urban areas adjacent to Wulie, Sanjia, and Shiyuetian Town, indicating a close relationship between the distribution of $NO_3$-type water and urbanization and other human activities in the study area. Furthermore, in the majority of the sampled groundwater, the concentration of $NO_3^-$ exceeded that of $SO_4^{2-}$, indicating a strong impact of human activities (such as untreated sewage, industrial wastewater, and agricultural fertilizers and pesticides) on the groundwater chemicals in the study area.

Analysis of the Gibbs diagram indicated that the groundwater chemical characteristics in the study area were mainly influenced by water–rock interactions. The water–rock interaction was dominantly affected by silicate weathering, and less affected by the weathering of carbonate rocks and evaporite rocks. Additionally, there was evidence of significant cationic exchange, leading to a decrease in $Ca^{2+}$ and $Mg^{2+}$ concentrations and an increase in $Na^+$ and $K^+$ concentrations in the water. The ion ratio analysis showed that $Na^+$ and $K^+$ mainly originated from the leaching of silicate rocks, while $Ca^{2+}$ and $Mg^{2+}$ primarily came from the dissolution of carbonate rocks. The presence of $NO_3^-$ was primarily related to human activities—particularly associated with agricultural activities and untreated domestic wastewater, rather than industrial activities. Considering these findings on groundwater pollution, appropriate prevention and remediation measures should be implemented in the study area.

The hydrochemical characteristics of groundwater are not only influenced by water–rock interactions but also undergo significant changes due to intensified human activities and the development of natural resources. To better understand the impact of human activities on the hydrochemical characteristics of groundwater in the study area, a comprehensive investigation should be conducted from various perspectives, such as spatial and temporal distributions and human-induced activities. This will provide reliable information for the development and sustainable utilization of groundwater resources in the study area.

**Author Contributions:** D.W., original draft preparation, methodology, writing; L.Z., L.P. and X.L., review and project administration; Y.Y. and Z.C., investigation; Z.C. and L.L., visualization and formal analysis. All authors have read and agreed to the published version of the manuscript.

**Funding:** This work was supported by the China Geological Survey Program, Land-Sea Coordination of Basic Geological Survey Results Integration and Expression of Key Technologies (DD20230416) and Ecological Restoration Support Survey Project of Changhua River Basin of Hainan Island (ZD20220209).

**Data Availability Statement:** The data presented in this study are available on request from the corresponding author.

**Conflicts of Interest:** The authors declare no conflict of interest.

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
