# Peer review of "Chemical Characteristics and Controlling Factors of Shallow Groundwater in the Lower Reaches of Changhua River Basin, Hainan Island, China"

_water, doi:10.3390/w15193508_

Round 1

Reviewer 1 Report

In the article entitled "Chemical Characteristics and Controlling Factors of Shallow Groundwater in the Lower Reaches of Changhua River Basin, Hainan Island, China", authors are trying to delineate the the chemical characteristics and the formation mechanism of the groundwater along an area of the Changhua River Basin. The manuscript, seems to be well-written and worth reading, however, some edits are needed as highlighted in the attached file.

Author Response

Dear reviewers

Re: Manuscript ID: water-2597342 and Title: Chemical Characteristics and Controlling Factors of Shallow Groundwater in the Lower Reaches of Changhua River Basin, Hainan Island, China

Thank you for taking the time to review our manuscript (water-2597342) and for providing your valuable comments. We have carefully gone through each comment and made the necessary corrections. As per your instructions, we have uploaded the revised manuscript for your review. Any additions to the text have been highlighted in blue. We appreciate your feedback and look forward to hearing your thoughts on the revised version.

Best regards,

Q1. The abstract is too long and needs to be shortened and improved.

Response: We are grateful for the suggestion. As suggested by the reviewer, we have shortened and improved the abstract.

Q2. Not only hydrogeochemical but also bio-physicochemical processes associated with organic matter presence. Please check these references and elaborate here on this diversity.

https://doi.0rg/l 0.1016/j.jappgeo.2013.08.011

https://doi.org/10.1080/01490451.2015.1049676

Response: We are grateful for the suggestion. To be more clear and by the reviewer's concerns, we have added a more detailed interpretation regarding the importance of the chemical composition of groundwater, and re-wrote the sentence in the revised manuscript as the following: “The chemical composition of groundwater is not only an important part of the hydrogeochemical cycle but also closely related to biophysical and chemical processes related to the presence of organic matter. Therefore, the study of groundwater chemistry is closely related to the ecological environment.” at the same time we cited relevant references as auxiliary explanations.

Q3. Please fix the scale(Figure 1. Location and geomorphological diagram of the study area)

Response: We acknowledge the comment and have made the necessary optimization and adjustments to Figure 1.

Q4. any justification for the sample numbers/uneven distribution? ( 3.1. Sample collection)

Response: Thank you for your attention to this matter. We have explained the distribution of sampling points in our article titled "Investigating the Hydrochemical Characteristics and Controlling Factors of the Lower Changhua River Basin in Hainan Island". Our study takes into account the hydrogeological conditions, geological and geomorphic features, as well as land use types in the area. We collected a total of 100 groundwater samples, comprising 51 pore groundwater samples and 49 fissure groundwater samples, which can represent the groundwater characteristics of the lower Changhua River basin in Hainan Island to the greatest extent possible. As a result, the sample size and distribution may be uneven. Thank you once again for your interest in our research.

Q5. Please unify the text font and format across the whole manuscript(4. Results and discussions)

Response: Thank you very much for your opinion. We have indented the first line of the first paragraph as per your comment.

Q6. Figure resolution is low. (Figure 3. Chemical types and spatial distribution of groundwater in the study area)

Response: We are grateful for the suggestion. We have improved the image's expression and resolution.

Q7. Please highlight what are (a), (b), (c) and (d) in the figure the caption(Figure 4; Figure 5; Figure 6; Figure 7)

Response: We appreciate your suggestion and have detailed the content of the image in the image caption above.

Q8. This table needs a reformat to avoid confusing the reader. (Table 3. )

Response: We are grateful for the suggestion. We have checked the format of Table 3 in detail. It may be due to an abnormality in the time when converting Word to PDF, so we have reformatted Table 3.

Reviewer 2 Report

The authors have done a comprehensive survey of shallow groundwater in the Lower Reaches of Changhua River Basin, Hainan Island in China. The data analysis of water samples found out that the groundwater is in a near-neutral to weakly alkaline condition. The percentage and various cation/anion ratio indicate that the groundwater in this region is mainly affected by human activity such as agricultural activities and domestic sewage, which provides the reliable guidance for the development and sustainability of groundwater utilization in the area.

Recommend publishing after minor revision.

Page 10: In the paragraph below the figure, Mg2+ should be Mg2+

Page 11: Figure 7 should be Figure 6.

Author Response

Dear reviewers

Re: Manuscript ID: water-2597342 and Title: Chemical Characteristics and Controlling Factors of Shallow Groundwater in the Lower Reaches of Changhua River Basin, Hainan Island, China

We feel great thanks for your professional review work on our article. As you are concerned, there are several problems that need to be addressed. According to your nice suggestions, we have made extensive corrections to our previous draft, the detailed corrections are listed below. Revisions in the text are shown using blue highlight.

We would love to thank you for allowing us to resubmit a revised copy of the manuscript and we highly appreciate your time and consideration.

Sincerely.

Q1. Page 10: In the paragraph below the figure, Mg2+ should be Mg2+

Q2. Figure 7 should be Figure 6.

Response:We apologize for the mistake caused by our oversight. We fully acknowledge and appreciate the feedback provided by the reviewers and have promptly made the necessary adjustments. Specifically, on page 10, we have corrected the mention of Mg2+ to Mg2+ in the relevant paragraph. Additionally, on page 11, we have corrected the reference to Figure 7 to Figure 6.

Reviewer 3 Report

The main aim of manuscript is research of chemical composition of shallow groundwater of Changhua River Basin, Hainan Island, China. Main processes and factors of chemical composition formation of ground water were conducted with using classical geochemical methods. The paper was written in well-structured manner. The text of the manuscript is sufficiently supplemented with figures and tables.

In my opinion, the article presents the results of a high-quality study and corresponds to the level of the journal Water. I would like to wish the authors to supplement the research by studying the trace element composition of groundwater in future studies. This would allow the authors to create a comprehensive environmental assessment of the area's natural waters.

In addition to this wish, I have a few minor comments:

I would recommend that authors include line numbers in their manuscripts. This makes the work of reviewers easier.

It would be great to revise the keywords. Current keywords repeat the title of the manuscript. This is not a good choice in terms of your article's visibility in search requests.

Should indicate the brands of devices in the methodology.

The chemical characteristics of groundwater are affected by natural factors such as precipitation, evaporation, surface water, and sedimentary environment, as well as human factors such as pollution and mining, which are the product of long-term interac-tion between groundwater and the surrounding environment [4-5].” Precipitation, evaporation and others are processes, are not they? However, in the same lane there are also surface waters. In my opinion, the name of the process is missing in front of “surface water”. This could probably be “infiltration” or “mixing with”.

Figure 2 almost repeats the fragment from Figure 1. It would be nice to combine these two pictures.

The first paragraph of the section “4. Results and discussions” has a font that is different from the rest of the text. It's worth unifying.

Author Response

Dear reviewers

Re: Manuscript ID: water-2597342 and Title: Chemical Characteristics and Controlling Factors of Shallow Groundwater in the Lower Reaches of Changhua River Basin, Hainan Island, China. We deeply appreciate your feedback and comments on our manuscript. Your input has been invaluable, and we have carefully reviewed your suggestions and made the necessary corrections.

As per your instructions, we have uploaded the revised version of our manuscript. The additions in the text have been highlighted in blue. We want to extend our gratitude for allowing us to resubmit the revised copy of our manuscript and for your time and consideration.

Thank you again for your valuable feedback and for helping us improve our manuscript.

Sincerely

Q1. I would like to wish the authors to supplement the research by studying the trace element composition of groundwater in future studies. This would allow the authors to create a comprehensive environmental assessment of the area's natural waters.

Response: We appreciate your suggestion and agree to include your additions in our follow-up work. Our next step is to conduct research on trace elements in the groundwater of the study area.

Q2. I would recommend that authors include line numbers in their manuscripts. This makes the work of reviewers easier.

Response: We apologize for our oversight and have added line numbers to the revised manuscript to make it easier for reviewers to read.

Q3. It would be great to revise the keywords. Current keywords repeat the title of the manuscript. This is not a good choice in terms of your article's visibility in search requests.

Response: We are very grateful for the suggestions on the selection of keywords in the article. The keywords have been revised in the revised manuscript, and the description is as follows: " Hainan Island; Hydrogeochemistry; rock weathering; Controlling factors; groundwater "

Q4. Should indicate the brands of devices in the methodology.

Response: We fully support your suggestion of including the brand of testing equipment in the method. However, some equipment with a long history has had their brand omitted. Therefore, not all equipment brands are indicated in the method. Nevertheless, all the brands in this study The source and scientific nature of the data can be obtained from the corresponding author.

Q5. “The chemical characteristics of groundwater are affected by natural factors such as precipitation, evaporation, surface water, and sedimentary environment, as well as human factors such as pollution and mining, which are the product of long-term interaction between groundwater and the surrounding environment [4-5].” Precipitation, evaporation and others are processes, are not they? However, in the same lane there are also surface waters. In my opinion, the name of the process is missing in front of “surface water”. This could probably be “infiltration” or “mixing with”.

Response: We are grateful for the suggestion. To be more clearly and in accordance with the reviewer's concerns, and re-wrote the sentence in the revised manuscript as follows: “The chemical characteristics of groundwater are affected by natural factors such as precipitation, evaporation, infiltration surface water, and sedimentary environment, as well as human factors such as pollution and mining, which are the product of long-term interaction between groundwater and the surrounding environment.”

Q6. Figure 2 almost repeats the fragment from Figure 1. It would be nice to combine these two pictures.

Response: We are grateful for the suggestion. We have re-modified the content of Figure 1 to make Figures 1 and 2 more helpful for explaining the article.

Q7. The first paragraph of the section “4. Results and discussions” has a font that is different from the rest of the text. It's worth unifying.

Response: We apologize for any inconvenience caused by our negligence. The necessary modifications have been made based on the review comment.

Round 2

Reviewer 1 Report

Authors have addressed all my comments.

Author Response

thank you